# How Does a Community Respond to Changes in Aircraft Noise? A Comparison of Two Surveys Conducted 11 Years Apart in Ho Chi Minh City

**DOI:** 10.3390/ijerph18084307

**Published:** 2021-04-19

**Authors:** Bach Lien Trieu, Thu Lan Nguyen, Yasuhiro Hiraguri, Makoto Morinaga, Takashi Morihara

**Affiliations:** 1Graduate School of Natural Science and Technology, Shimane University, 1060 Nishikawatsu-cho, Matsue, Shimane 690-8504, Japan; trieulien0903@gmail.com; 2Faculty of Architecture, Kindai University, 3-4-1 Kowakae, Higashiosaka City, Osaka 577-8502, Japan; hiraguri@arch.kindai.ac.jp; 3Defense Structure Improvement Foundation, 15-9 Yotsuya-Honshio-cho, Shinjuku-ku, Tokyo 160-0003, Japan; d4-morinaga@bsk-z.or.jp; 4Department of Architecture, National Institute of Technology, Ishikawa College, Tsubata, Kahoku-gun, Ishikawa 929-0932, Japan; morihara@ishikawa-nct.ac.jp

**Keywords:** changed noise environment, aircraft noise, annoyance, health effects

## Abstract

There have been many arguments about findings of an increase in noise annoyance over time and a recommendation of stricter limits on aircraft noise levels to protect the health of residents around airports. It is crucial to examine if the established exposure–response relationship is suitable for designing future aircraft noise regulations. This study was focused on identifying changes in response to noise over time by comparing community responses from two surveys conducted in 2008 and 2019 at Tân Sơn Nhất (TSN) international airport. Annoyance was found to significantly reduce in 2019 compared to 2008; however, changes in sleep quality were relatively small. Unexpectedly, a gradual increase in the annoyance due to aircraft noise was not found. Results of multiple regression analysis indicated that differences in the reaction of the residents to noise in the two studies were significantly attributed to nonacoustic factors. Noise sensitivity and dissatisfaction with the living environment (e.g., inconvenience in accessing workplace) considerably affect noise annoyance, whereas noise sensitivity, age, and dissatisfaction with the green environment of living areas affect sleep quality. These findings suggest the fulfillment of desired living environment as effective measures for mitigating noise impacts on residents in the vicinity of busy airports.

## 1. Introduction

It was decided in the European Ministerial Conference on Environment and Health held in Parma in 2010 to develop new guidelines on noise. To this end, the World Health Organization (WHO) conducted a systematic review [1,2,3,4] of the effects of environmental noise and announced their results in the Environmental Noise Guidelines for the European Region (2018) [5]. These guidelines strongly recommended reducing aircraft noise levels to 45 dB *L*_den_ (day-evening-night-weighted sound pressure level) and 40 dB *L*_night_ (night-time equivalent continuous sound pressure level) to protect the health of residents around airports. However, although the recommended values were derived using data obtained globally, few data points were obtained from Asia (in particular, from Asian developing countries). Further, European and American studies report that the reaction of people to aircraft noise increases in severity every year [6]. Recent research on the change caused by the opening of a new terminal building at Noi Bai Airport at the end of 2014 suggests that responses obtained several years later are higher than those obtained before the change occurred under the same noise level. However, the effect due to operational change decreases, as observed in the follow-up study regarding annoyance, but it remains the same with regard to insomnia [7,8]. Further studies in developing countries are thus required to confirm these findings.

The Tân Sơn Nhất (TSN) international airport—located inside a very dense residential area of Ho Chi Minh City, the most active metropolitan area in Vietnam—is the largest airport in Vietnam, with over 250,000 movements, and it served approximately 40 million passengers in 2018 [9].

This paper compares the community responses from two surveys conducted in 2008 and 2019 around the TSN. The 2019 study acts as a follow-up examination of the community’s response to noise after 11 years by surveying the same areas as the 2008 study [10]. The number of flights at present is 3.3 times greater than that in 2008. This research project aims to answer the following questions:(1)Is there a secular change in the community reaction owing to the increase/decrease in exposure to aircraft noise?(2)Are the WHO guidelines applicable to developing Asian countries?

This paper is divided into four parts. After the introduction, Section 2 is concerned with the materials and methodology used. Section 3 presents the findings of the research, focusing on changes in annoyance and sleep effects corresponding to changes in noise exposure and residential and nonacoustic factors between 2008 and 2019. Section 4 discusses the effects of nonacoustic factors on noise exposure–response relationships, identifies attitudes towards the noise source of Ho Chi Minh residents, and analyzes the implication of the findings on the environmental quality standard for aircraft noise. The limitation of the study is presented in the remaining part of the discussion. Finally, the conclusion gives a summary and critique of the findings.

## 2. Materials and Methods

### 2.1. Survey Sites

#### 2.1.1. Survey 2008

Ten residential areas were selected around TSN airport (Sites A1–A10), eight sites under the landing and takeoff paths of the aircraft and the two others at the north and south of the runway, respectively (see Figure 1). The site selection was intended to reflect aircraft noise exposure covering locations at various distances from and in directions relative to the airport. Further, because this survey was intended to investigate aircraft noise both as a single and as a combined source, all sites except Sites A9 and A10 were selected from residential areas that had roads passing through them. The houses facing the roads were selected for the combined noise survey, and those set back from the road were selected for the single-aircraft noise survey. Because the main purpose of our study was to determine the impact of aircraft noise at different ranges on residents in the noise-affected area and to identify residential and nonacoustic factors that can moderate these effects, only data from the single-aircraft noise survey were used.

#### 2.1.2. Survey 2019

In the follow-up survey, a total of 10 sites (Sites B1–B10) near the sites of the 2008 survey were investigated.

### 2.2. Socio-Acoustic and Health Surveys

#### 2.2.1. Survey 2008

Community responses to aircraft noise were investigated around the TSN airport between August and September 2008. Because the TSN airport was operated in a relatively stable manner throughout the year, this survey period was selected to unify with other surveys conducted in Vietnam so that the obtained data can represent a situation of one year around the time of the study. The surveys were conducted via face-to-face interviews during the daytime on weekends. The interviewers visited and collected responses from all of the residences in the selected study area. Because the response rate was quite high, nonrespondent analysis was not performed. In both surveys, the composition of the interviewees in each household was adjusted to have the same rate of demographic factors as that in the Vietnam Census. In particular, to ensure a balance between males and females and generations, fathers, mothers, and other adults in the family were selected for the survey. The status of the interview participants in terms of whether they worked for the airport was not investigated. Therefore, some areas of the 2009 and 2019 surveys may not be fully comparable regarding the degree of benefit from the airport. The survey areas were selected to reflect the exposure situation from minimum to loudest aircraft noise levels, so they can be considered as representative of the population around the airport in terms of noise exposure.

The design of the questionnaire followed the Technical Specification ISO/TS 15666 [11]. The questionnaire not only focused on noise but also on various components of the living environment. The content of the questionnaire contained queries on housing, neighborhood environment, noise annoyance, interference with daily activities, sensitivity, attitude towards transportation, and sociodemographic items.

#### 2.2.2. Survey 2019

Face-to-face interviews were conducted in August 2019 using a questionnaire including the items related to general annoyance and effects on sleep from the 2008 survey. Further, data on the health status of residents were collected to evaluate the effects of aircraft noise around the TSN airport. We focused on analyzing and comparing similar data between the two surveys. The primary outcomes of noise impact considered in this study were annoyance and self-reported sleep disturbances. Table 1 lists the questions and scales used to assess them in both surveys.

Annoyance and sleep effects are the most widely used measures of the human response to noise. The standardized annoyance question and the 11-point numeric scale used in this study are as recommended by the International Commission on Biological Effects of Noise (ICBEN) [12]. In both surveys, the aircraft noise-induced annoyance was represented by the percentage of respondents who were highly annoyed (%HA): percentage of respondents who chose 8, 9, or 10 out of the 11-point numeric scale (0–10).

In the 2019 survey, the percentage of insomnia was considered as the frequency of effects on sleep, as proposed in previous studies [13,14,15], and it was used as an indicator of the effect that flight operation had on sleep during the night. The questionnaire on insomnia symptoms was not used in the 2008 survey, and therefore, to compare the effect on sleep between the two studies, we used data relating to sleep quality measured by two similar-content questions used in the two surveys. In the 2008 study, sleep quality was assessed by the question, “What is the status of your daily sleep?”. The respondents were asked to respond to each item on a five-point scale ((1) Extremely good; (2) Good; (3) Neutral; (4) Bad; (5) Extremely bad). In the 2019 survey, the question was worded, “During the past four weeks, how would you rate the quality of your sleep?”. There were four alternative responses: (1) Very good; (2) Fairly good; (3) Fairly bad; (4) Very bad. The sleep effect of noise was represented by the percentage of respondents who had low sleep quality (%LSQ) defined by the percentage of respondents who chose “4. Bad” and “5. Extremely bad” categories in the 2008 survey and those who chose “3. Fairly bad” and “4. Very bad” categories in the 2019 study. Because the evaluation scales are different, we can only compare the trends of the outcomes.

### 2.3. Noise Exposure Data

#### 2.3.1. Survey 2008

The predicted values and data required for prediction such as flight route, runway use, flight operation data, and airplane performance could not be obtained in the 2008 survey. Therefore, field measurement values were used to estimate the noise exposure, *L*_den_ and *L*_night_, in this survey, which within the same site was considered equal. Aircraft noise exposure was measured every 1 s for seven successive days, from September 22 to 29, 2008, using sound level meters (NL-21 and NL-22, RION, Tokyo, Japan). Microphones covered with all-weather wind screens were positioned on the rooftops of the highest houses in the areas (1.5 m above the roofs and at least 1 m away from any other reflecting surface). The information such as flight arrival and departure times were obtained from the airport office.

#### 2.3.2. Survey 2019

Noise measurement and flight route data collection for the estimation of noise exposures around the airport were conducted simultaneously. To check the accuracy of the noise estimation, the noise was measured for one week (August 4–11) by applying the same method as in the 2008 survey. *L*_den_ and *L*_night_ were estimated from noise contour maps calculated using the Integrated Noise Model 7.0 (INM) [16]. Flight route data were collected from 5–9 August 2019, with an Automatic Dependent Surveillance-Broadcast (ADS-B) installed in the airport office building at a location with good visibility to obtain flight route information in every one-second interval. Because of the prevailing wind direction, almost all take-offs and landings at the TSN are to the west throughout the year. The three-day flight route data and flight log data, including the number of operations by aircraft type and takeoff and landing provided by TSN, were used to calculate the noise contour maps.

The estimation was based on the flight data logged for the entire survey period of one week. The flight data log was compared with the seasonal average traffic to ascertain that the estimated period was representative of the noise scenario. Predictive calculations were performed for three days, and the average value was used as the representative value of *L*_den_ and *L*_night_. According to the flight logs, the average arrivals and departures at TSN in a day were counted and classified into day, evening, and night periods.

The relationship between the obtained noise exposure and the reaction of the residents was clarified for the data of the 2019 survey, and the change in the reaction of the residents over the 11-year span was examined by comparing it with the exposure–response relationship obtained in 2008.

In the maps shown in Figure 1 and Figure 2, the dots represent the installation locations of the sound level meters, while the zoned areas represent the area of the questionnaire surveys. In the 2019 survey, the measurements were not performed at Site B4 because no appropriate house for noise measurement installation was available. Noise measurements at sites B3 and B8 failed because of errors in data storage. Therefore, the dots are not shown for these sites in the map, and no comparison was conducted between estimated and measured values at these sites.

Noise exposure obtained from field measurement was used in the analysis for the 2008 survey, with one exposure value per survey site, whereas calculated values were used instead of the measurements in the analysis for the 2019 survey, with the exact values at the residents’ addresses.

### 2.4. Statistical Analysis

Multiple logistic regression analysis was conducted using data from both surveys to determine the change in the relationships between noise exposure and community response when considering the moderation effects of residential and nonacoustic factors. The possible effect of changes in the response of the residents to noise between the 2008 and 2019 surveys was investigated by adding a survey factor (Survey 2008: 0 and Survey 2019: 1). The analysis aimed to determine correlations between *L*_den_ and %HA and between *L*_night_ and %LSQ, as adjusted by other nonacoustic factors. First, all factors and the survey factors were applied to the model as influencing factors. Then, the model constructed with only the variables that significantly affected the prevalence of annoyance or LSQ was estimated. Generalized R-Square (*R*^2^) and area under the receiver operating characteristics (ROC) curve (AUC) were derived for each prediction model. *R*^2^ is the proportion of variance in the dependent variable (HA and LSQ) that can be predicted from the independent variables (noise exposure, residential, and nonacoustic factors). AUC measures how well predictions are ranked, and ranges from “0” to “1”. A model whose predictions are 100% wrong has an AUC of 0.0, whereas one whose predictions are 100% correct has an AUC of 1.0. All statistical analyses were performed using JMP 13.0.0 software (SAS Institute Inc., Cary, NC, USA).

## 3. Results

### 3.1. Demographic Data of the Respondents

A total of 880 and 502 responses were obtained in the 2008 and 2019 surveys, respectively. The demographic data of the survey respondents are summarized in Table 2. A higher response rate was achieved in the 2008 survey. In both surveys, the proportion of female respondents was slightly higher. Respondents aged over 60 years accounted for 11% and 18% of the total number of respondents in the 2008 and 2019 surveys, respectively. These obtained proportions are consistent with the characteristics of the young population (less than 60 years) of Vietnam. The proportion of employed respondents in the follow-up survey was higher than that in the 2008 survey.

### 3.2. Increase in Number of Flights and Noise Levels

The number of operated flights and passengers at the TSN airport has increased significantly over the past 11 years. Table 3 summarizes the average number of daily flights operated by the TSN during the two survey periods. The number of night flight events accounted for approximately 18.3% of the total number of flights in 2019, while this number was 13.8% in the 2008 survey. The increase in night-time flights is attributed to the rapid growth of low-cost carriers, which prefer operating at night (22:00–6:00) to save costs; this trend seems to reduce the flight components during the day and evening. The same pattern was observed at Hanoi Noi Bai Airport [8].

Table 4 shows the noise levels estimated using INM compared to the noise level data derived from the field measurements conducted at the corresponding sites in the 2019 survey. The root means square (RMS) differences between the predicted estimates and the corresponding measured values are 1.9 and 2.4 for *L*_den_ and *L*_night_, respectively. These discrepancies are well accepted considering that the 1–3 dB difference in the noise level is barely noticeable to the human ear [17]. The correspondence between the predicted and measured values verified the validity of the noise estimation for the 2019 survey. This indicated that the measured values for 2008 and the predicted values for 2019 could be used for the comparison.

Table 5 shows the average noise levels obtained during each survey period; the noise levels in the 2008 survey represent measured values, and those in the 2019 survey are the predicted values. *L*_den_ obtained at Sites A1–A10 ranged from 53–71 dB in 2008, while *L*_den_ ranged from 63–81 dB at the noise-exposed sites B1–B10 in 2019. These ranges are 45–62 dB in 2008 and 55–74 in 2019 with noise exposure at night. Among the 10 investigated sites, the measurement points at Site A10 of the 2008 survey and B10 of the 2019 survey coincided, and both *L*_den_ and *L*_night_ were found to increase by 7 dB.

The TSN airport has two parallel runways in the east–west direction (07L–25R and 07R–25L). The runway was used quite differently in 2019 than in 2008. In 2008, the airport used the southern runway (07R–25L) located close to the civil airport terminal, while the northern runway (07L–25R) served as a military airbase. In 2019, the military airbase was relocated to another place, and both runways were used for civilian aircrafts to meet the high density of flight flow at the airport. The three sites on the landing side in 2019 (Sites B1–B3) were shifted to the north compared to the corresponding three sites in 2008 (A1–A3) with the same distance to the 25R runway end. The difference in noise levels between these sites ranges between 5–12 dB, which reflects the increase in the flight numbers and the more frequent use of the northern runway. The noise exposure in the 2008 survey included noise released by military aircrafts, while there is only noise from civil flights in the 2019 survey.

### 3.3. Changes in General Annoyance and Sleep Effects

As summarized in Table 6, %HA ranged between 0 and 52% in the 2008 survey (corresponding to a 53–71 dB range of *L*_den_). Meanwhile, this range in 2019 is 0–18% (corresponding to a 63–81 dB range of *L*_den_). The highest %HA was at Site 5 in the 2008 survey, while in the 2019 survey, it was at Site B6. The %HA was only 3% at the noisiest site in the 2019 survey (Site B5), which is located nearest to the 25R runway end on the landing side and exposed to 81 dB (*L*_den_). Despite having to live in a noisier environment than before, the residents around the TSN airport seem more tolerant to the noise.

The percentage of low sleep quality in the 2008 survey ranged between 2% (Site A10, 53 dB *L*_night_) to 27% (Site A5, 62 dB *L*_night_). This range is 4% (Site B10, 60 dB *L*_night_) to 35% (Site B6, 66 dB *L*_night_) for the 2019 survey. Comparing the data for Sites A10 and B10, which are two coincident survey areas, the %LSQ was observed to increase from 2% to 4%. However, the %LSQ at Site B5 in the 2019 survey was low, at only 15%, with a high *L*_night_ of 74 dB. The highest %LSQ of the 2019 survey was 35% at Site B6, an area of 66 dB (*L*_night_). This result is consistent with the high %HA obtained at this site. Although residents around the TSN airport seem to be more tolerant of noise than that in the 2008 survey, the trend for reported sleep quality is different. This finding indicates that the effect of noise at TSN airport on the sleep quality of residents requires more attention from airport operators. The correlation coefficient between %HA and %LSQ in the 2008 survey, *r* = 0.725 (*p* = 0.0177), indicates a strong positive linear relationship. The correlation coefficient between %HA and %LSQ in the 2019 survey, *r* = 0.289 (*p* = 0.4176), indicates a weak positive linear relationship.

Table 7 summarizes the percentage and number of highly annoyed respondents in the two surveys at different noise exposure level ranges. The *p*-value derived by the Wald test determines whether *L*_den_ can be used to predict or correlate with %HA. The *p*-value shows that *L*_den_ was significantly associated with %HA in the 2008 survey at the <0.0001 level and in the 2019 survey at the <0.01 level. That is, higher noise levels increased the possibility of being highly annoyed in both surveys.

A comparison was drawn using the data on %LSQ at different ranges of *L*_night_ (Table 8). The *p*-value derived by the Wald test shows that *L*_night_ was significantly associated with %LSQ in the 2008 survey. This association was not observed in the 2019 survey data (*p* = 0.3974).

A logistic regression analysis was used to establish an exposure–response relationship for each study. Figure 3 shows a comparison of (a) *L*_den_–%HA and (b) *L*_night_–%LSQ relationships for the two studies. The *L*_den_–%HA relationship of the 2019 survey was lower than that of the 2008 survey, while the *L*_night_–%LSQ relationships of the 2019 survey almost coincided with that of the 2008 survey. However, the 2019 curve is much flatter. The curve for the 2019 study can be considered an extension of the 2008 curve.

Relationships obtained in this study were compared to those established in the Position Paper on European Union (EU) Noise Indicators [18] and the Environmental Noise Guidelines for the European Region by the WHO [5]. The %HA was obtained from the top 27–28% on the evaluation scale. The exposure–response relationship of both surveys was found to be lower than the relationship established in the WHO guidelines. The exposure–response relationship established in 2008 was closer to the relationship established in the EU position paper. The relationship of the 2019 survey was lower than those established in the WHO guidelines, which indicated the tolerance of the residents to aircraft noise around the TSN airport.

### 3.4. Influence of Residential and Nonacoustic Factors

#### 3.4.1. Residential and Nonacoustic Factors

Nonacoustic factors influence the reported aircraft noise annoyance and activity disturbance as significantly as the noise exposure level [19,20,21,22]. The data on the residential and nonacoustic factors of the respondents investigated in both surveys are listed in Table 9; they include residential, personal, and attitudinal factors. All these factors are assumed to modify the reactions of the respondents to noise. The average data for these factors obtained from each category are summarized in Table 9. Chi-squared tests of independence were performed by comparing these factors for the two surveys.

A significant difference was observed in most categories, except housing type, house width, sex, sensitivity to cold, vibration, chemicals, odors, and job components. The negative evaluation of residential areas regarding green, street scenery, view, quietness, work convenience, education convenience, health care convenience, daily life service convenience, and transport convenience—defined by the percent of bad and extremely bad responses—was lower in the 2019 survey. That is, the respondents in the 2019 survey were considerably more satisfied with their living areas than those in the 2008 survey. This change is consistent with the fact that with the positive change in the economy, the living amenities of the residents around the TSN airport improved, including the increased use of air conditioners. This improvement was indicated by a decrease in the percentage of open windows.

#### 3.4.2. Multiple Logistic Models for Annoyance and Low Sleep Quality

As noted in Section 2.4, multiple logistic regression models were constructed with only the variables that significantly affected the prevalence of annoyance or LSQ selected from factors listed in Table 9. Table 10 and Table 11 show the results of the analysis. Significant associations were found between *L*_den_ and annoyance (*p* < 0.0001) and between *L*_night_ and LSQ (*p* = 0.0008). The survey factor adjusted by the other nonacoustic factors significantly affected the prevalence of annoyance at the <0.0001 level and the LSQ at the <0.001 level. The variable representing the interaction of noise exposure and survey factor, survey factor * *L*_den_ (Table 10) and survey factor * *L*_night_ (Table 11), had a significant effect on annoyance and LSQ, respectively. It is worth noting that the coefficient of the interaction between noise level (*L*_den_ or *L*_night_) and survey factor is negative in both models. It indicated that the effect of survey factor decreased when noise exposure increased, and vice versa.

The evaluation of work convenience and noise sensitivity had a significant effect on the prevalence of annoyance. Age, evaluation of the green environment, and noise sensitivity had significant effects on the prevalence of LSQ. The odds of residents above 60 for LSQ are 2.497 times of that under 60. In other words, general noise annoyance is influenced by work convenience assessments, while sleep quality is influenced by the age and the green surroundings around the house. These results suggest that residents may be less likely to be annoyed if they find their living areas convenient for work. The sleep quality of residents in noisy areas can be improved if they are satisfied with the green surroundings around the house.

The coefficient of the survey factor is negative in the model of annoyance but positive in the models of LSQ. With the adjustment of the survey factor, the rate of negative response to the noise around the TSN airport decreased in terms of general annoyance but increased in terms of low sleep quality. The odds that the resident was highly annoyed in the 2019 survey compared to those of the 2008 survey was 0.260. In other words, the odds that a resident was highly annoyed by noise in the 2008 survey was 3.850 times the odds in the 2019 survey. On the contrary, the LSQ model showed that the odds of having bad sleep quality in 2019 were 2.177 times the odds in 2008.

The strength of the association between nonacoustic factors and annoyance or LSQ was quantified using the analysis method proposed by Michaud et al. [23]. The stepwise regression was carried out with the base model including only noise level. Next, the model included noise level and survey factor; then, each nonacoustic factor was added to the model in order to make *R*^2^ of the model increase gradually.

In the annoyance model, as shown in Table 12, survey factor was the first factor to enter the multiple logistic regression model, and the corresponding R^2^ in the base models increased from approximately 6.5% to 18.7%. Sensitivity factor added 6.2% to R^2^. The other variables added the remaining 2% to *R*^2^. In the LSQ model, as shown in Table 13, noise sensitivity was the first factor to enter the multiple logistic regression model, and the corresponding *R*^2^ in the base models increased from approximately 2.9% to 6.6%. Survey factor was added at Step 4 and did not relate to LSQ as strongly as the noise sensitivity and age factors.

The exposure–response relationships summarized in Table 10 and Table 11 are presented in the form of graphs in Figure 4. Figure 4a compares the *L*_den_–%HA relationships, while Figure 4b compares the *L*_night_–%LSQ relationships in the 2008 and 2019 surveys, respectively, as adjusted by the moderators analyzed in Table 10 and Table 11. The exposure-response relationships adjusted by other factors, shown in Figure 4a,b, are significantly different from those considered independently, which are shown in Figure 3a,b. The response to aircraft noise of the residents in Ho Chi Minh City changed in the past 11 years and was influenced by nonacoustic factors as important as noise exposure.

The adjusted curve drawn for the 2019 survey is lower than that for the 2008 survey in the case of the *L*_den_–%HA relationship. The adjusted curve drawn for the 2019 survey is slightly higher than that of the 2008 survey in the case of the *L*_night_–%LSQ relationship. Further, the prevalence of low sleep quality in the two surveys was measured using different scales. The %LSQ of the 2008 survey was obtained from the top two categories of the five-point scale or the top 40%, while that of the 2019 survey was from the top two categories of the four-point scale or the top 50%. This difference may have contributed to the gap between the relationships. The elevated prevalence of low sleep quality in the 2019 survey adds to concerns that a gradual increase in noise exposure and enhanced flight operation at night-time around the TSN airport may negatively affect the resident’s sleep quality.

However, under the influence of nonacoustic factors such as satisfaction with the green environment and the convenience of transportation to the workplace, the sleep quality and the general noise annoyance are respectively reduced, which indicates the potential to improve the quality of life in noisy environments through other environmental measures.

## 4. Discussion

### 4.1. Effects of Nonacoustic Factors on Noise Exposure–Response Relationships

Noise sensitivity was found to be a significant factor affecting the prevalence of annoyance and low sleep quality. This finding is consistent with that of previous studies, which defined self-reported noise sensitivity as a nonacoustic factor that significantly influences noise exposure–response relationships [21,22,23,24]. Further, dissatisfaction with the living environment in terms of inconvenience to work affects noise annoyance. Thus, satisfaction with convenient access to the workplace reduced the negative response for Ho Chi Minh City.

According to the results of a 2019 survey on the population and housing census of Vietnam [25], the housing and living conditions in Vietnam improved over the last ten years, especially in the urban areas. The number of households that now owned modern living facilities increased compared to those in the 2009 census. Remarkably, the percentage of households in Vietnam equipped with air conditioners increased by 25.5% (2009: 5.9%, 2019: 31.4%). This rate was 52.9% in 2019 (urban areas: 57.0, rural areas: 37.5) for the households in Ho Chi Minh City. Although no such data was available for this city in the 2009 census, according to the Vietnam Household Living Standard Survey in 2010 [26], this rate for households in the Southeast region of Vietnam having half the population of Ho Chi Minh City was 11.8% in 2008 and 14.5% in 2010. The increased number of air-conditioner-equipped houses is directly related to the decreased window-opening frequency. This change may improve the sound insulation performance of windows, thereby indirectly making the house more insulated to noise and contributing to a reduction in noise annoyance. This moderation effect should be examined further because annoyance also depends on outdoor exposure because people often stay outdoors.

Nonacoustic factors such as noise sensitivity, age, and dissatisfaction with the green environment of living areas were found to influence the sleep quality of residents in Ho Chi Minh City at a higher significance level compared to night-time noise exposure. This result is consistent with a study on more than 259,000 Australians, which found that people living in greener neighborhoods reported a lower risk of short sleep [27]. This result suggests that more green space within the neighborhood environment can help ease the negative sleep effect of the increased noise scenario. This finding is inconsistent with recent findings of Schäffer et al. in their Swiss study that increasing residential green is associated with decreased road traffic and railway noise annoyance whereas it is strongly linked to increased aircraft noise annoyance [28]. This difference is explained by the intrusiveness of aircraft noise, which contrasts expectations of green spaces. The incongruence of sound and landscape is unfavorably perceived. Therefore, the adverse reaction to aircraft annoyance was often observed in the countryside. However, the TSN airport is in the center of a densely populated city; therefore, it has very different environmental settings from those presented in the Swiss study.

The perception of an increase in noise may differ in a rapidly changing economic context and under changes in other aspects of living conditions. For example, the proportion of households using air conditioners has doubled in 10 years, and this is accompanied by a surge in aviation and road traffic. This change in home furnishings and the satisfaction of the people with living convenience makes them more accepting of noise as an inevitable result of economic growth. Therefore, their response was not as high as the expected levels according to the corresponding noise exposure–response relationship. However, the need for tranquility when sleeping at night is constant regardless of the economic or cultural context, and it is independent of the fulfillment of other living conditions. That is, an increase in noise levels at night might cause an overreaction in the people living around the TSN airport despite their tolerance to noise during the daytime.

A crucial factor that can affect the response to aircraft noise in Ho Chi Minh City is the airport and air-transport attitude factor. The TSN airport was initially planned to have an area of 3600 ha. However, after the Vietnam war (post 1975), the land around the airport was divided for people working there. Although there has been a plan to remove houses around the TSN to build a five-meter-wide safety corridor around the airport since 2002, implementing this plan was impossible because people continued to invade and extend their residences to the extent that houses were adjacent to airport fences. Currently, only approximately 1500 ha of the airport planned area remains. Therefore, it could be assumed that the low annoyance level at B5 is associated with the people’s perception of the benefits of living near the airport.

Considering that Site B5 has different characteristics from other sites, we re-analyzed the dataset excluding it. However, the results did not change significantly when compared to the presented results. As the population around the airport grows and accommodation demands increase, residents do not complain about noise because they feel it is more important to live in this area than to be affected by the noise. The fact that the residents have adapted and are ready to endure the noise has resulted in less annoying self-reported data. A previous study on the environmental attitudes of the residents near Noi Bai Airport in northern Vietnam found that people can accept living near a noisy airport for its benefits despite its environmental impact [29].

Because the attitude factor was not investigated in the 2019 survey, we used data from the most recent study conducted in 2020 with households living in the same areas as the 2019 survey to compare the data with the attitude data of the 2008 survey. In both the 2008 and 2020 surveys, the attitude toward air transport was assessed in terms of (a) the use frequency, (b) benefit to society, and (c) safety (see Appendix A in the Appendix A for the attitude data comparison of the two surveys). In 2020, there was a tendency to improve the attitude toward air transportation compared to 2008. Further responses show an agreement with more frequent use, benefit to society, and safe air transport found in 2020 than in 2008. It appears that more positive attitudes toward air transport have mitigated the negative response to noise from aviation operations.

The results of the multiple logistic regression analyses indicated that differences in the reaction of the Ho Chi Minh City residents to noise were significantly attributed to nonacoustic factors such as age, noise sensitivity, convenience in accessing the workplace, and the green environment of living areas (Table 10 and Table 12). Among them, the survey factor was found to have the most significant effect on the annoyance level. The survey factor might include whether or not military aircrafts are in operation, the spread of air-conditioner use, and attitudes toward air transport. In this study, it is unknown to what extent these factors contribute to the significant effect of the survey factor. Yokoshima et al. reported that annoyance caused by military aircraft noise was much higher than that caused by civil aircrafts in Japan [30]. Nguyen et al. reported that attitudes toward the frequency of air-transport use significantly affected noise annoyance [10]. Therefore, although the impact of the spread of air-conditioner use is unknown, it could be assumed that the elimination of military aircraft operations and the improvement of attitudes toward a noise source have significantly contributed to reducing the annoyance of people living around TSN airport. The survey factor also contains methodological differences between the 2008 and 2019 surveys, as shown in Appendix A, which affected the outcomes of the surveys.

### 4.2. Change in Aircraft Annoyance and Implications for the Environmental Quality Standard for Aircraft Noise

The significant influence of the survey factor in the multiple regression analysis confirmed the difference in the residents’ reactions to the noise between the two surveys. The reaction to aircraft noise in 2019 was lower in terms of annoyance but higher in terms of low sleep quality compared to the reaction in the 2008 study. The relationship of annoyance in the 2019 survey was lower than that established in the EU position paper and the WHO guidelines. The findings of this study suggest that the degree of reaction to the increase in aircraft noise differs between annoyance and sleep effects. The community response to aircraft noise in Ho Chi Minh City in terms of annoyance in 2019 was considerably lower than that in 2008 despite the exposure to high noise levels from 63–81 dB (*L*_den_) and 3.3 times more flight events compared to those in 2008. In contrast, the response in 2019 was slightly higher than that in 2008 at the same noise level in terms of sleep effect.

The result of decreased annoyance in the 2019 survey compared with that of the 2008 survey in Ho Chi Minh City is opposite to the that observed in recent studies that reported aircraft noise annoyance increased in Europe. Babish et al. assessed noise annoyances caused by aircraft and road traffic noise from subjects living in the vicinity of six major European airports using the 11-point ICBEN scale; the same scale we used in our study [31]. The exposure–response curve for aircraft noise was higher than that of the EU standard curve. Janssen et al. found a significant increase in annoyance over the years at a given level of aircraft noise exposure using a database of 34 airports [32]. Reports by both Babish et al. [31] and Janssen et al. [32] suggest that the EU standard curve for aircraft noise should be modified. This conclusion is not in line with this study’s finding that the excess response did not occur with an annoyance reaction but with a sleep effect in the TSN case.

The relationship between aircraft noise exposure and annoyance of the WHO’s guideline was found to be higher than that in the EU position paper, as shown in Figure 3. Recommendations in the guideline were based on systematic reviews of evidence from individual studies wherein the effect of aircraft noise on the self-reported annoyance and sleep outcomes was measured [1,2]. Stricter limits recommended in the guidelines were questioned for their validity. Gjestland criticized the inclusion of the results of surveys conducted at airports that underwent a change scenario, such as after the opening of a new runway or an increase in the number of flights [33]. Such scenarios result in a higher prevalence of annoyance or insomnia. The results of the re-analysis of 61 surveys on road traffic noise conducted over the past 45 years in Gjestland’s study indicate that the annoyance caused by road traffic noise is stable across the entire period [34].

Another criticism by Gjestland is that the WHO’s revised recommendations are based on a limited selection of publications, in which studies relied on nonstandard questionnaires, respondent selection, and definitions of annoyance prevalence rates that over-estimated annoyance [35]. His re-analysis indicated that no meaningful changes in the prevalence rates of high annoyance with aircraft noise have occurred. However, Brink asserted that Gjestland’s review lacks a sound meta-analysis driven by a clearly formulated research question, including the disclosure of the criteria for study selection and description of data extraction; therefore, it could not confirm definitively whether aircraft noise annoyance has increased over the past decades [36].

Another reason for the decreased annoyance at a given noise exposure might be that the noise exposure increased strongly around the TSN airport, whereas in Europe, noise exposure remained stable or even decreased over time, which may be changing the expectations of residents. Gjestland and Gelderblom examined the community tolerance level values (CTL) for 32 aircraft noise surveys concerning the yearly number of aircraft movements [37]. Their study found that around low-rate-of-change airports like European airports, the prevalence of highly annoyed residents increases with the number of movements. However, the same tendency cannot be found for high-rate-of-change airports like the TSN airport. At airports experiencing large changes in their operational patterns, the annoyance assessment is most likely dominated by other nonacoustic factors, and the effect of the number of movements seems to be absent or masked. This conclusion does not strongly support, but is not contrary to, our study’s findings.

### 4.3. Limitations

This study has a few limitations. First, our surveys were cross-sectional, as opposed to cohort studies that are preferable for this research. To compare changes in community response, obtaining data from the same respondents over time (cohort studies) is preferable. Although it was impossible to track the same respondents, we intended to survey the same areas as in the 2008 survey. However, after 11 years, with changes in residential scenarios, we could not ensure that the areas surveyed in 2019 and in 2008 coincided. Therefore, instead of comparing each pair of survey areas, we compared the exposure–response relationship between the two periods with all datasets. This does not significantly affect the results because this study only reported trends in changes of response levels of communities living near TSN airport without targeting a specific resident.

Second, the noise exposure data from the 2008 survey were obtained from field measurements, while those from the 2019 survey were estimated from the noise map. Although the predicted and measured values were almost consistent, obtaining noise exposure data from the noise maps throughout the entire study would have been more favorable.

Third, regarding the operation of military aircrafts in the 2008 survey, it would be more favorable to clearly separate the contributions of civil and military aircrafts to the noise level, thereby identifying the impact levels between the two types of noise sources. Unfortunately, we only have total aircraft noise data. Therefore, we cannot ascertain if the higher annoyance in 2008 was associated with the presence of military aircraft noise.

Fourth, all data on public reactions were derived from face-to-face interviews conducted in the order of father, mother, and adults besides parents in each house. Although this method seems to be almost similar to random sampling and the demographic distribution almost matches the Vietnamese census, it would be better to apply the random sampling method. However, random sampling based on residence registration may not be possible at present in Vietnam. A resident register is a government database containing information on the current residence of persons. Vietnam does not have a transparent residence registration database. Most houses, especially in densely populated areas, usually have their registered address only including the street name and lane without having a residence number or address specific enough for an interviewer to be able to find an individual household.

## 5. Conclusions

We analyzed the data of 2008 and 2019 aircraft noise surveys in Ho Chi Minh City and compared changes in noise annoyance and sleep quality based on the results of both surveys. Annoyance was significantly reduced in 2019 compared to that in 2008; however, changes in sleep quality were relatively small. This study demonstrates a contradictory tendency compared to that presented in recent studies, which report that aircraft noise annoyance increases over time. The decline in annoyance in the 2019 survey was found to be related to increased satisfaction with the convenience of accessing the workplace. The other cause is attributed to the increased number of households equipped with air conditioners, which indirectly reduced indoor noise exposure because the residents could close windows more frequently. Satisfaction with the green environment of living areas was found to lower the rate of low sleep quality. The positive air-transport attitudes of the residents were also found to be an important factor that contributed to minimizing aircraft noise annoyance in Ho Chi Minh City. These findings can help policymakers, aviation authorities, and environmental managers to design effective measures for mitigating noise impacts on residents in the vicinity of busy airports.

## Figures and Tables

**Figure 1 ijerph-18-04307-f001:**
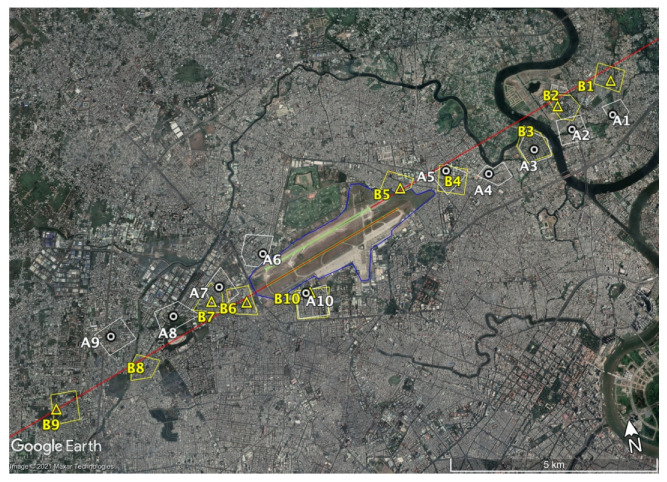
Map of surveyed sites in 2008 (A1–A10) and 2019 (B1–B10).

**Figure 2 ijerph-18-04307-f002:**
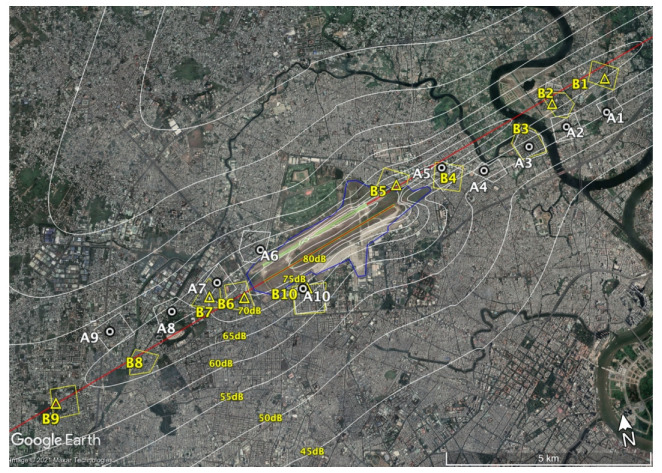
Noise map of 2019.

**Figure 3 ijerph-18-04307-f003:**
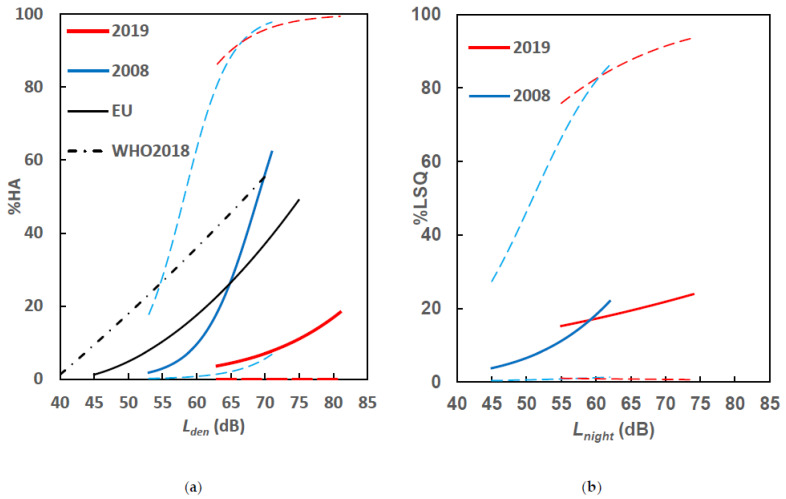
Comparison of (**a**) *L_den_*–%HA and (**b**) *L_night_*–%LSQ relationships with 95% confidence interval between the 2008 and 2019 surveys.

**Figure 4 ijerph-18-04307-f004:**
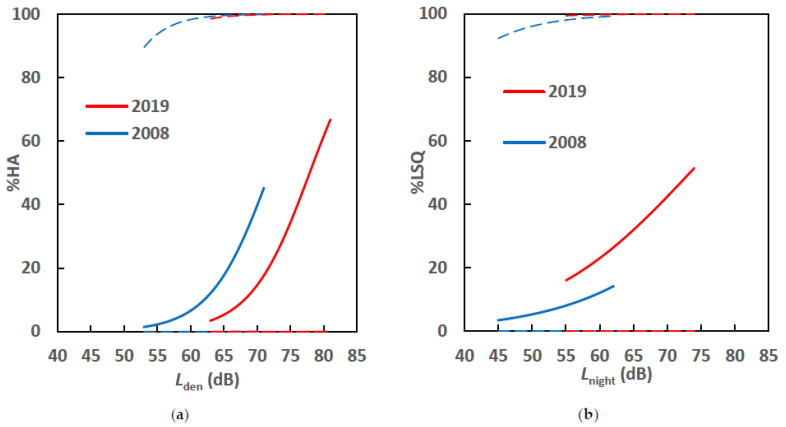
Comparison of (**a**) *L*_den_–%HA and (**b**) *L*_night_–%LSQ relationships adjusted by nonacoustic variables including sex, age, green and convenience evaluation of the residential areas, and noise sensitivity.

**Table 1 ijerph-18-04307-t001:** Outcome: Annoyance and sleep disturbance questions used in the surveys.

Survey	Annoyance Questions	Sleep Disturbance Questions
2008	Thinking about the last 12 months or so, what number from 0 to 10 best shows how much you are bothered, disturbed, or annoyed by aircraft noise? 11point scale used from 0 (not at all) to 10 (extremely) (HA ^a^:8, 9, 10)	Q17. In daily life, when an airplane passes by, to what degree are you disturbed in the following cases: Q17_6. When it makes it difficult for you to fall asleep? Q17_7. When you are awakened from your sleep? Not at all; Slightly; Moderately; Very; Extremely. (HSD ^b^: Very, Extremely) Q20. How is the status of your daily sleep? Extremely good; Good; Neutral; Bad; Extremely bad. (LSQ ^c^: Bad, Extremely bad)
2019	Same	Q8. How often do you have trouble getting to sleep or staying asleep? 1. Often; 2. Sometimes; 3. Almost never Q9. How many hours of sleep do you usually get at night? 6 h or less; 7 h; 8 h; 9 h or more Q10. During the past 4 weeks, how would you rate the quality of your sleep overall? 1. Very good; 2. Fairly good; 3. Fairly bad; 4. Very bad. Q11. Please answer this question concerning your sleep: Q11.1. Do you have any trouble with your sleep? (1) No; (2) Yes. If you answered “Yes” to the above question, please choose appropriate numbers for each item. (1) Difficult to fall asleep. (2) When awakened during the night, it is difficult to sleep again. (3) Awakened early in the morning. (4) Do not feel as having slept well the next morning. (5) Sleepy during daytime and cannot work well. (6) Others Occasionally; Once or twice a week; More than three times a week Q11.2. If you have trouble with your sleep, do you think that it is due to the aircraft noise? (1) No; (2) Yes

^a^ Highly annoyed. ^b^ Highly sleep disturbed. ^c^ Low sleep quality.

**Table 2 ijerph-18-04307-t002:** Demographic data of the respondents in both surveys.

Items		Surveys		Vietnamese Census (2019) *
		2008	2019	
Number of respondents		880	502	
Response rate (%)		88	60	
Sex (%)	Male	47	46	50
	Female	53	54	50
Age (%)	20–50 years	89	82	88
	≥60 years	11	18	12
Occupation (%)	Employment	45	54	74 ^a^ 55 ^b^
	Student, homemaker, retired, unemployed	55	46	26 ^a^ 45 ^b^

(*): General Statistics Office in Vietnam, “Statistical Date” http://www.gso.gov.vn/default_en.aspx?tabid=491 (accessed on 22 February 2021). ^a^ calculated in >15-year-olds population. ^b^ calculated in all population.

**Table 3 ijerph-18-04307-t003:** Average number of aircraft noise events.

Time Period	Operation Modes	Surveys	
		2008	2019
Day (6:00–18:00)	Arrival	67	214
	Departure	82	244
	Total	149	458
Evening (18:00–22:00)	Arrival	28	73
	Departure	16	64
	Total	44	137
Night (22:00–6:00)	Arrival	17	77
	Departure	14	56
	Total	31	133
All day	Arrival	112	364
	Departure	112	364
	Total	224	728

**Table 4 ijerph-18-04307-t004:** Comparison of noise levels obtained from noise map and field measurement in the 2019 survey.

Noise Levels		Survey Sites												RMS ^c^
		1	2	3	4	5	6	7	8	9	10	11	12	
*L* _den_ ^a^	Estimated	64.3	65.4	65.5	62.8	81.3	74.2	69.8	66.1	63.7	67.4	47.3	45.0	1.9
	Measured	65.6	67.9	-	-	78.9	73.7	70.4	-	64.6	63.8	45.8	44.2	
*L* _night_ ^b^	Estimated	56.6	57.6	57.7	55	73.6	66.2	61.9	58.3	55.9	59.6	39.8	37.5	2.4
	Measured	58.5	60.7	-	-	71.6	66.9	63.1	-	56.8	55.4	36.6	35.9	

^a^ Day-evening-night-weighted sound pressure level. ^b^ Night-time equivalent continuous sound pressure level. ^c^ Root means square.

**Table 5 ijerph-18-04307-t005:** *L*_den_^a^, *L*_night_^b^, and their changes from the 2008 survey to 2019 survey.

2008 Survey			2019 Survey		
Site	*L* _den_ ^a^	*L* _night_ ^b^	Site	*L* _den_ ^a^	*L* _night_ ^b^
Site A1	59	52	Site B1	64	57
Site A2	53	45	Site B2	65	58
Site A3	55	48	Site B3	66	58
Site A4	57	49	Site B4	63	55
Site A5	71	62	Site B5	81	74
Site A6	64	56	Site B6	74	66
Site A7	66	58	Site B7	70	62
Site A8	62	55	Site B8	66	58
Site A9	62	54	Site B9	64	56
Site A10	60	53	Site B10	67	60

^a^ Day-evening-night-weighted sound pressure level. ^b^ Night-time equivalent continuous sound pressure level.

**Table 6 ijerph-18-04307-t006:** Percentage of highly annoyed (%HA) and percentage of low sleep quality (%LSQ).

2008 Survey				2019 Survey			
Site	%HA ^a^	%LSQ ^b^	No. of Responses	Site	%HA ^a^	%LSQ ^b^	No. of Responses
Site A1	5	7	85	Site B1	0	14	48
Site A2	0	8	86	Site B2	7	12	41
Site A3	7	3	90	Site B3	0	27	31
Site A4	9	8	90	Site B4	2	18	49
Site A5	52	27	90	Site B5	3	15	33
Site A6	49	11	83	Site B6	18	35	49
Site A7	34	12	90	Site B7	13	10	48
Site A8	11	9	88	Site B8	6	12	32
Site A9	3	13	89	Site B9	0	22	45
Site A10	1	2	89	Site B10	2	4	33

^a^ Percentage of respondents who were highly annoyed. ^b^ Percentage of respondents who had low sleep quality.

**Table 7 ijerph-18-04307-t007:** Comparison of percentage of highly annoyed (%HA) at different noise level ranges of the 2008 and 2019 surveys.

		Noise Level Ranges L_den_ ^a^ (dB)				*p*-Value
		<60	60–65	65–70	>70	
2008 survey	%HA	5.2	15.5	34.4	52.2	<0.0001
	Response number/N	17/330	53/341	31/90	47/90	
2019 survey	%HA		0.7	6.1	12.2	0.0082
	Response number/N		1/142	12/197	10/82	

^a^ Day-evening-night-weighted sound pressure level.

**Table 8 ijerph-18-04307-t008:** Comparison of percentage of low sleep quality (%LSQ) at different noise level ranges of the 2008 and 2019 surveys.

		Noise Level Ranges L_night_ ^a^ (dB)					*p*-Value
		<50	50–55	55–60	60–65	>65	
2008 survey	%LSQ	6.5	7.8	11.6	26.7		<0.0001
	Response number	17/260	27/345	20/172	24/90		
2019 survey	%LSQ			15.2	10.4	26.8	0.3974 (n.s)
	Response number			45/297	5/48	22/82	

^a^ Night-time equivalent continuous sound pressure level.

**Table 9 ijerph-18-04307-t009:** Data of residential and nonacoustic factors of the respondents investigated in the two surveys.

Factors	Categories	2008 Survey	2019 Survey	*p*-Value
Residential Factors				
Housing type	Self-owning	65.6 (576/878)	64.9 (321/495)	0.8743
Floor Area/Width of house	≤50 m^2^	53.2 (462/868)	59.1 (269/455)	0.2632
Housing structure	1. Wooden 2. Brick 3. Prefabricated 4. Reinforced concrete 5. Reinforced concrete with brick wall 6. Others	1.8 (16/871) 39.6 (345/871) 1.4 (12/871) 16.9 (147/871) 39.4 (343/871) 0.9 (8/871)	2.1 (7/337) 14.5 (49/337) 0.3 (1/337) 44.8 (151/337) 34.4 (116/337) 3.9 (13/337)	<0.0001
Number of glass layers in living room windows and doors	1. More than 3 layers 2. 2 layers 3. 1 layer 4. Others (the window has no glass)	0.1 (1/866) 7.5 (65/866) 71.7 (621/866) 20.7 (179/866)	2.7 (13/490) 18.2 (89/490) 75.3 (369/490) 3.9 (19/490)	<0.0001
Type of frame of living room windows and doors	1. Aluminum frame 2. Wooden frame 3. Plastic frame 4. Others	32.2 (276/858) 25.8 (221/858) 1.9 (16/858) 40.2 (345/858)	31.7 (156/492) 14.2 (70/492) 1.4 (7/492) 52.6 (259/492)	0.0061
Number of glass layers in bedroom windows and doors	1. More than 3 layers 2. 2 layers 3. 1 layer 4. Others (the window has no glass)	0.5 (4/850) 6.8 (58/850) 68.6 (583/850) 24.1 (205/850)	1.2 (6/488) 13.5 (66/488) 77.7 (379/488) 7.6 (37/488)	<0.0001
Type of frame of bedroom windows and doors	1. Aluminum frame 2. Wooden frame 3. Plastic frame 4. Others	27.5 (234/850) 31.8 (270/850) 4.5 (38/850) 36.2 (308/850)	37.3 (181/485) 20.0 (97/485) 2.7 (13/485) 40.0 (194/485)	0.0115 *
Personal and attitudinal factors				
Sex	Male	47.1 (411/872)	46.2 (229/496)	0.8467
Age	≥60 years old	10.5 (91/867)	18.1 (90/498)	0.0490 *
Residence length Residential area preference and quality (% Bad and Extremely bad)	≤5 years 1 Green 2 Street sceneries 3 View 4 Quietness 5 Work convenience 6 Education convenience 7 Health care convenience 8 Daily life service convenience 9 Transport convenience	55.2 (478/866) 21.2 (185/873) 24.5 (214/874) 13.8 (119/865) 15.1 (131/870) 9.8 (85/863) 22.5 (195/868) 26.0 (226/869) 24.1 (209/868) 16.4 (142/866)	41.7 (204/489) 12.3 (60/487) 7.9 (38/483) 8.0 (39/485) 9.0 (43/478) 3.8 (18/475) 1.9 (9/478) 3.4 (16/477) 1.3 (6/477) 4.4 (21/478)	0.0063 ** 0.0070 ** <0.0001 0.0275 * 0.0341 * 0.0015 ** <0.0001 <0.0001 <0.0001 <0.0001
Opening of bedroom windows (% Often and Always)	1. Dry season 2. Rainy season	34.8 (301/865) 24.4 (210/860)	31.2 (140/449) 17.9 (81/452)	0.0157 * 0.0005 ***
Sensitivity(% Very and Extremely	1. Cold 2. Heat 3. Noise 4. Vibration 5. Chemicals 6. Odors 7. Dust, pollen, polluted air	3.0 (26/862) 24.2 (209/862) 26.0 (224/860) 5.7 (49/854) 7.8 (67/856) 14.2 (122/861) 12.0 (103/860)	2.9 (14/480) 15.6 (75/482) 16.1 (78/483) 8.5 (41/482) 5.4 (26/480) 8.8 (42/480) 6.7 (32/481)	0.9528 0.0166 * 0.0072 ** 0.3210 0.2869 0.0551 0.0326 *
Job	1. Employed 2. Student 3. Homemaker 4. Retired 5. Unemployed	45.3 (392/865) 10.2 (88/865) 15.6 (135/865) 13.9 (120/865) 15.0 (130/865)	53.6 (266/496) 9.3 (46/496) 13.1 (65/496) 9.7 (48/496) 14.3 (71/496)	0.4471
Number of hours staying at home	1. Under 8 h 2. From 8 to 15 h 3. Above 15 h	8.2 (64/784) 27.0 (212/784) 64.8 (508/784)	30.6 (149/487) 36.6 (178/487) 32.6 (159/487)	<0.0001

* *p* < 0.05, ** *p* < 0.01, *** *p* < 0.001.

**Table 10 ijerph-18-04307-t010:** Multiple logistic regression for annoyance (HA) (Generalized *R*^2^: 0.2815; AUC: 0.856).

Item	Category	Estimate	Std Error	*p*-Value	Odds Ratio	Lower 95%	Upper 95%
Intercept		−16.509	1.624	<0.0001			
*L* _den_ ^a^		0.224	0.025	<0.0001	1.250 ^b^	1.313 ^b^	0.800 ^b^
Survey factor	2008 survey				1		
	2019 survey	−1.348	0.361	0.0002	0.260	0.128	0.527
*L*_den_^a^ × Survey factor		−0.187	0.050	0.0002			
Sex	Male				1		
	Female	0.100	0.199	0.6156	1.105	0.748	1.633
Age	≤60 years				1		
	>60 years	0.622	0.304	0.0406	1.864	1.027	3.381
Green	Satisfied				1		
	Dissatisfied	0.330	0.244	0.1753	1.392	0.863	2.244
Work convenience	Satisfied				1		
	Dissatisfied	1.084	0.279	0.0001	2.956	1.710	5.110
Noise sensitivity	Insensitive				1		
	Sensitive	1.527	0.200	<0.0001	4.604	3.109	6.820

^a^ Night-time equivalent continuous sound pressure level. ^b^ Odds ratio in 1 dB change.

**Table 11 ijerph-18-04307-t011:** Multiple logistic regression for low sleep quality (LSQ) (Generalized *R*^2^: 0.1054; AUC: 0.733).

Item	Category	Estimate	Std Error	*p*-Value	Odds Ratio	Lower 95%	Upper 95%
Intercept		−7.963	1.487	<0.0001			
*L* _night_ ^a^		0.090	0.027	0.0008	1.095 ^b^	1.154 ^b^	0.914 ^b^
Survey factor	2008 survey				1		
	2019 survey	0.778	0.227	0.0006	2.177	1.394	3.400
*L*_night_^a^ × survey factor		−0.098	0.037	0.0078			
Sex	Male				1		
	Female	0.230	0.185	0.2130	1.259	0.876	1.809
Age	≤60 years				1		
	>60 years	0.928	0.232	<0.0001	2.529	1.605	3.987
Green	Satisfied				1		
	Dissatisfied	0.708	0.221	0.0014	2.030	1.316	3.133
Work convenience	Satisfied				1		
	Dissatisfied	0.064	0.344	0.8529	1.066	0.543	2.093
Noise sensitivity	Insensitive				1		
	Sensitive	1.190	0.204	<0.0001	3.288	2.206	4.901

^a^ Night-time equivalent continuous sound pressure level. ^b^ Odds ratio in 1 dB change.

**Table 12 ijerph-18-04307-t012:** Stepwise regression analysis for annoyance.

Variable	Category	Odds Ratio (95% CI)	*p*-Value	Order of Entry into Model: *R*^2^ at Each Step
*L* _den_ ^a^	Continuous	1.129 (1.095, 1.164) ^b^	<0.0001	Base: 0.0647
Survey factor	2019/2008	0.074 (0.043, 0.129)	<0.0001	Step 1: 0.1871
Sensitivity	Sensitive/Insensitive	4.500 (3.083, 6.567)	<0.0001	Step 2: 0.2487
Work convenience	Dissatisfied/Satisfied	2.646 (1.554, 4.507)	0.0003	Step 3: 0.2641
Age	>60 years/≤60 years	1.750 (0.976, 3.135)	0.0669	Step 4: 0.2676
Green	Dissatisfied/Satisfied	1.201 (0.757, 1.904)	0.4368	Step 5: 0.2682
Sex	Female/Male	1.077 (0.734, 1.581)	0.7046	Step 6: 0.2674

^a^ Day-evening-night-weighted sound pressure level. ^b^ Odds ratio in 1 dB change and 95% confidence interval (95% CI).

**Table 13 ijerph-18-04307-t013:** Stepwise regression analysis for LSQ.

Variable	Category	Odds Ratio (95% CI)	*p*-Value	Order of Entry into Model: *R*^2^ at Each Step
*L* _night_ ^a^	Continuous	1.087 (1.053, 1.121) ^b^	<0.0001	Base: 0.0290
Sensitivity	Sensitive/Insensitive	2.916 (2.030, 4.187)	<0.0001	Step 1: 0.0663
Age	>60 years/≤60 years	2.437 (1.571, 3.781)	<0.0001	Step 2: 0.0808
Green	Dissatisfied/Satisfied	1.752 (1.154, 2.658)	0.0084	Step 3: 0.0873
Survey factor	2019/2008	1.681 (1.087, 2.602)	0.0197	Step 4: 0.0932
Sex	Female/Male	1.256 (0.876, 1.801)	0.2156	Step 5: 0.0951
Work convenience	Dissatisfied/Satisfied	1.045 (0.534, 2.043)	0.8987	Step 6: 0.0976

^a^ Night-time equivalent continuous sound pressure level. ^b^ Odds ratio in 1 dB change and 95% confidence interval (95% CI).

## Data Availability

Data sharing not applicable.

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
