# Peer review of "How Does a Community Respond to Changes in Aircraft Noise? A Comparison of Two Surveys Conducted 11 Years Apart in Ho Chi Minh City"

_ijerph, 2021, doi:10.3390/ijerph18084307_

Round 1

Reviewer 1 Report

This is a very serious paper, well crafted and documented. Congratulations to the authors!

It provides an in debt analysis of the reactions of the neighbors regarding the noise from the airport as well as reports results from noise level measurements at several locations.

Authors had focused mainly in the areas of annoyance and sleep interference. Since there are so many elements that affect comfort, they have chosen two of the most important for their analysis.

The time difference between both tests – 11 years – is large enough as to draw significant conclusions. During this period the airport as well as the surrounding had experienced tremendous changes, especially with he increase of flights and the density of the population.

The large amount of data collected by the authors has been craftily analyzed as to arrive to valid conclusions as well as to suggest future researches.

This reviewer appreciated authors finding real-life reasons for people to dismiss negative effects, such as been close to work or having easy transportation to downtown.

It reminded him a situation he observed where a neighbor to a noisy factory denied any noise problem, because the factory was providing him with water for free…

In summary, a great paper that does not need corrections.

Reviewer 2 Report

In this paper, the results of two field surveys (one of 2008 and the other of 2019) on high annoyance (%HA) and low sleep quality (LSQ) around TSN airport in Vietnam are compared. Annoyance strongly decreased while LSQ slightly increased between 2008 and 2019. A range of factors explaining these findings are explored.

The study is interesting and an important contribution to the discussions on annoyance changes to aircraft noise over time. The manuscript is well structured and well written, it was a pleasure to read. Nevertheless, I have several comments on the manuscript, as detailed below. Given that the comments are considered in the revision, I recommend the manuscript to be accepted for publication.

Specific comments:

Introduction:

L42-45: "… and Brown et al. … developed countries [4]." These sentences do not really fit in here, and I suggest removing them.

L55-59 and L64-67: These lines give information on a potential follow up study on the community reactions to aircraft noise. As it is not the topic of the current study, I suggest removing them for sake of brevity.

Materials and methods:

L98: "control areas" for what? Also, these areas are not listed in Table 5. Were they included in the survey? If not, could this information be removed?

L122: "sleep disturbance": I would specify as "self-reported sleep disturbance"

L127-129: "recommended by ICBEN": ICBEN does not recommend this for the 11-pointe scale: See ref [13]: "No recommendation is offered here for a definition of "high'' annoyance using the more abstract numeric scale, as the respondents' answers do not provide a clear basis for a division." Nevertheless, the 8-10 points are common practice.

The methods section should be complemented with a "Statistical Analysis" section to introduce the logistic regression analysis and details of the analysis

General point regarding the surveys: As the study compares the 2008 and 2019 surveys, it would be helpful to have a table listing the similarities and differences in the two surveys (e.g., exposure assessment methods, choice of survey areas etc.). This is even more important as your statistical analysis revealed the "survey factor" to be the most important variable, indicating that there is a number of methodological differences between the two surveys. Such a table could be presented as supplementary materials.

Details on recruiting: How were the participants recruited? Did you just go into the areas and asked people to participate? Did you do a non-respondent analysis to check for potential bias? How representative are the relatively small survey sites around the airport for the whole population around the airport? Are the areas of the 2009 and 2019 surveys fully comparable (e.g., did they to the same degree benefit from the airport, e.g. regarding employment)?

Survey 2019: Please state here that you used the calculations instead of the measurements in the analysis. Did you use one exposure value per survey site or the exact values at the residents' addresses?

Results:

General: Did you do a correlation analysis between %HA and LSQ? As both are subjective measures, I would expect them to be at least partly correlated.

L287-289: "relationships of the 2019 survey almost coincided with that of the 2008 survey … " However, the 2019 curve is much flatter.

Figs. 3 and 4: Please also show 95% confidence intervals.

L302-304: "nonacoustic factors … residential": "residential" is not purely non-acoustic, but also covers an acoustic component: E.g. the better insulated the houses, the quieter it is inside.

Table 9: I cannot find the values for sensitivity.

L321-323 (and other instances): This should be presented in a separate "Statistical Analysis" section, giving more details there

Table 10: Please define generalized R2 and AUC.

Why did you not use a quadratic expression for age? See Van Gerven et al. 2009. Annoyance from environmental noise across the lifespan. J. Acoust. Soc. Am. 126, 187-194.

Interesting that satisfaction with green decreased aircraft noise annoyance. In a recent study, increased aircraft noise annoyance was found in greener residential areas (while annoyance to road traffic noise was decreased). See Schäffer et al. 2020. Residential green is associated with reduced annoyance to road traffic and railway noise but increased annoyance to aircraft noise exposure. Paper No. 105885. Environ. Int. 143.

The exposure-response curves between 2008 and 2019 show distinctly different slopes. Why did you not include an interaction Survey factor x Lden and Survey factor x Lnight?

L360-361: "exposure–response relationships adjusted by other factors are significantly different from those considered independently": Is that so? To my knowledge, adjustment for other factors narrows the confidence intervals, but does not change the shape of the curve, given that the factors are adjusted to the mean values of the study sample. Of course, here you set the curves to the 2008 sample values, so I expect the 2019 curves to be modified.

L362: "influenced by nonacoustic factors as important as noise exposure": This could be quantified using R2 of models step-wise including additional factors. See e.g. Table II of Michaudc 2016. Personal and situational variables associated with wind turbine noise annoyance. J. Acoust. Soc. Am. 139, 1455–1466.

Header Fig. 4: "sites classified with ΔLden and ΔLnight": What does that mean?

Discussion:

L402-405: " improve the sound insulation … reduction in noise annoyance." I partly agree, but annoyance will also depend on outdoor exposure, as people also stay outdoors.

Section 4.2: "Attitude" is also a non-acoustic factor. I suggest shortening this section and merging it with Section 4.1. L444-453: If you include another survey, it should be documented in the methods section. Fig. 5 should be presented in the results section, or as supplementary material, as it does not really fit the main 2019 survey.

Section 4.3:

Title: "Change in Aircraft Annoyance and Implications for Aircraft Noise Policy": I do not see a discussion on the implication for noise policy.

The "survey factor" should be discussed a bit more. I assume it contains also methodological differences between the 2008 and 2019 surveys, which may have affected the results?

Could another reason for the decreased annoyance at a given noise exposure be that the noise exposure increased strongly around TSN airport, while in Europe, noise exposure remained stable or even decreased over time, which may be changing the expectations of residents?

L487: Ref [31] is the same as Ref [1] and should be removed.

L488-494: If you cite Gjestland, you should also mention his (quite similar) criticism in JASA and, even more importantly, the comment by Brink on Gjestland's review:

1) Gjestland, 2020. Recent World Health Organization regulatory recommendations are not supported by existing evidence. J. Acoust. Soc. Am. 148, 511-517

2) Brink, 2020. J. Acoust. Soc. Am. 148, 3397–3398: Comment on “Recent World Health Organization regulatory recommendations are not supported by existing evidence” [J. Acoust. Soc. Am. 148, 511–517 (2020)](L)

L495-508: These lines fit better to Section 4.1, and the considerations about military aircraft could be mentioned in Section 4.4.
